# Prediction of Enzyme Function Based on Three Parallel Deep CNN and Amino Acid Mutation

**DOI:** 10.3390/ijms20112845

**Published:** 2019-06-11

**Authors:** Ruibo Gao, Mengmeng Wang, Jiaoyan Zhou, Yuhang Fu, Meng Liang, Dongliang Guo, Junlan Nie

**Affiliations:** School of Information Science and Engineering, Yanshan University, Qinhuangdao 066004, Hebei, China; grb664425@126.com (R.G.); mmw1218601537@163.com (M.W.); fjiaoyan121@163.com (J.Z.); fuyuhang_fly@163.com (Y.F.); liangmengxa@163.com (M.L.); dongliang1005@163.com (D.G.)

**Keywords:** enzyme function prediction, DCNN, amino acid sequence, mutation information

## Abstract

During the past decade, due to the number of proteins in PDB database being increased gradually, traditional methods cannot better understand the function of newly discovered enzymes in chemical reactions. Computational models and protein feature representation for predicting enzymatic function are more important. Most of existing methods for predicting enzymatic function have used protein geometric structure or protein sequence alone. In this paper, the functions of enzymes are predicted from many-sided biological information including sequence information and structure information. Firstly, we extract the mutation information from amino acids sequence by the position scoring matrix and express structure information with amino acids distance and angle. Then, we use histogram to show the extracted sequence and structural features respectively. Meanwhile, we establish a network model of three parallel Deep Convolutional Neural Networks (DCNN) to learn three features of enzyme for function prediction simultaneously, and the outputs are fused through two different architectures. Finally, The proposed model was investigated on a large dataset of 43,843 enzymes from the PDB and achieved 92.34% correct classification when sequence information is considered, demonstrating an improvement compared with the previous result.

## 1. Introduction

As the volume of protein databases has increased and new protein families have been discovered [1], protein function prediction is becoming more and more important since it allows estimating the properties of novel proteins according to the group to which they are predicted to belong.

Protein function is determined by protein structure and amino acid sequence. Protein structure, e.g., the 3D structure of a protein is a very good predictor of protein function [2]. Although structure relates to amino acid sequence, additional information can be extracted directly from the amino acid sequences [3]. For example, sequence homology is vital when considering the field of protein function prediction. Protein sequences are homologous if they have a common ancestral sequence [4]. The inheritance through homology is a common and accessible approach to function prediction because proteins with similar sequences frequently carry out similar functions [5]. In evolution, Mutations in amino acids on protein sequences may lead to changes in homology. The degree of difficulty in amino acid mutation is called amino acid conservatism. In recent years, the position scoring matrix has been mainly used to indicate the robustness of amino acids. PSSM has been widely used in the field of bioinformatics, such as nucleotide binding site prediction [6,7], and has made significant progress. Le et al. [8] proposed a method based on PSSM profiles and biochemical properties to identify the category of transport proteins from their molecular function, which provided a powerful model for prediction. Furthermore, Le et al. [9,10] respectively studied Rab protein molecular functions classification and SNARE proteins identification by combining PSSM profiles with deep CNN framework, which achieved an evident improvement in all the typical measurement metrics for protein functions prediction, but the model still need better input all of the PSSM information into deep CNN to avoid missing important features. Enzyme is a type of special protein and the assessment of similarities between amino acid sequences is usually performed by sequence alignment. The computing method of protein function prediction can be used to fill the gap between sequence data and the unknown properties of these proteins.

Protein function prediction is usually treated as a multi-label classification problem. Researchers have tried different methods in the last few decades for this problem [11,12]. For example, the first widely used method for protein function prediction is the Basic Local Alignment Search Tool (BLAST) [13], which finds the remote homologous and use these homologous proteins’ function information to predict the function of the query sequence, the most challenging method is to directly use protein sequence without any other resource for protein function prediction [14], but protein feature cannot be fully represented when sequence information is only used. However, the machine learning techniques are usually used to predict protein function from scratch. Currently most machine learning methods use features generated from the protein sequences for model training, and then the model classifies a number of function categories of input sequences [15,16]. For example, most methods apply support vector machines (SVM) [17], bayes classifiers [18], classification trees [19,20], and k-nearest neighbor [21]. Whereas, the prediction methods based on structural information are proposed by Borro et al. [22] and Amidi et al. [23,24]. These methods extract features from only sequences or structure.

In the past few years, Deep learning has been increasingly used to tackle many challenging problems in a wide variety of fields, such as image classification and protein function prediction [25,26,27,28]. The main advantage of deep learning techniques is the automatic exploitation of features and tuning of performance in a seamless fashion, which simplifies the conventional image analysis pipelines. CNN has recently been used for protein secondary structure prediction [29]. Spencer et al. [30] made use of the position-specific scoring matrix and deep learning network architectures to predict secondary structure of Ab initio protein, and the model did not appreciably advance the field of secondary structure prediction, but it did achieve several favorable results. Whereas Li et al. [29] proposed a method based on CNN and ensemble learning to predict protein secondary structure prediction. Though it was a significant step toward the theoretical limit for the prediction accuracy, the model was simple since it had only 2 base classifiers. Also, Lin et al. [31] proposed a deep CNN architecture based on protein sequences to predict protein properties, which achieved better results than whole sequence-based approaches like the GSN, LSTM, but the feature extraction was a bottleneck. Renzhi Cao et al. [32] built a neural machine translation model based on recurrent neural networks to extract useful information from protein sequences. The model showed better performance, but the accuracy and the reliability should be improved in future. Evangelia et al. [33] presented novel shape features representing protein structure based on a deep convolutional neural network to predict protein function, which was a considerable improvement over the accuracy when only structural information was incorporated, but extracted features did not capture the topological properties of the 3D structure. Le et al. [34] presented a 2D CNN constructed from position specific scoring matrix profiles to classify motor proteins. The approach achieved an improvement compared with traditional machine learning techniques and help biologist to discover new motor proteins, however, it still bears some limitations and there remain some possible approaches for improving this type of problem. So we hope to establish a comprehensive representation of protein features and use a novel deep learning model to train features.

In order to better understand the features that the neural network learns, we adopted a visualization pattern. Visualization is intended to provide a qualitative and easy to interpret indication of the features that are driving the CNN model’s output. Proteins are macromolecules consisting of chains of amino acid, and the sequence is commonly referred to as the primary structure of a protein and determines its native conformation. Understanding protein function helps to reveal the fundamentals of biochemical processes in living cells [35]. Because it is important in biology, a detailed overview of visualization methods for biomolecules has been presented by Kozl et al. [36]. Recently, a vast number of analysis and visualization techniques have been developed and applied to protein. Byska et al. [35] proposed a novel method for the visually interactive exploration of protein tunnels. Watanabe et al. [37] presented protein-protein interactions visualization system. Kayikci et al. [38] developed multiple representations for visualization and analysis of non-covalent contacts at different scales of organization. Therefore, we give a brief introduction to the training process and results.

This paper is organized as follows: We detailedly explain the experimental results in Section 2. In Section 3, we mainly introduce the method used in the experiment and feature maps. Finally, we provide some discussions and conclusions in Section 4 and Section 5.

## 2. Experiment and Results

The dataset about proteins (n = 43,843) was retrieved from the PDB (http://www.rcsb.org/) database. The enzymes are divided into 6 primary categories: oxidoreductases (EC1), transferases(EC2), hydrolases (EC3), lyases (EC4), isomerases (EC5), ligases (EC6). These enzymes are single class, and enzymes that perform multiple reactions and are associated with multiple enzymatic functions were excluded. What’s more, the enzymes of small sequence length or homology bias were are not considered. In the pre-process step of data, we first read the amino acid sequences in each pdb file of proteins online, then the scoring matrix of the protein was derived by PSI-BLAST.

We use the 80% of data as the training set and the rest 20% as the test set. The number of samples of per class is shown in Table 1. The training set not only were used to learn the parameters of the network, as well as the parameters of the classifiers used during fusion. Once the network was trained, all parameters were determined. When the test set is entered into the network, we can determine the likelihood of belonging to each class, which were introduced into the trained classifier for final prediction.

The convolutional layer used neurons with a receptive field of size 5 for the first two layers and 2 for the third layer. The stride (specifying the sliding of the filter) was always 1. The number of filters was 20, 50 and 500 for the three layers, respectively, and the learning rate is 0.001. The batch size was selected according to the dimensionality of the input (1000 for Architecture 1 and 100 for Architecture 2). The number of epochs was adjusted to the rate of convergence for each architecture, e.g., the epochs was 300 in Architecture 1 and the epochs was 150 for Architecture 2.

In this experiment, we use the proposed model with three feature sets $X_{A}$, $X_{D}$, $X_{L}$ (ADL-DCNN) compared with the model of two features $X_{A}$, $X_{D}$ (AD-CNN [33]). Because we employ two architectures (Architecture 1 and Architecture 2) of CNN to train and test the same data, respectively, when the training feature sets are the same, the result of different architectures can be compared each other. Likewise, in the same architecture, we can compare the predictions results of the model ADL-DCNN and AD-CNN.

In Table 1, the red figures show the classification accuracy for each enzyme class, and the last row indicates the overall prediction accuracy. The second column of Table 1 denotes the number of samples per class. The last four columns express the comparison of the accuracy for two architectures. In the same dataset, the accuracy of ADL-DCNN in Architecture 2 and Architecture 1 is 92.34% and 90.01%, respectively. Obviously, the performance of the proposed model ADL-DCNN is better than that of the model AD-CNN. In the same model, due to the multi-channel of Architecture 2, the prediction accuracy is higher than that of Architecture 1.

Based on Table 1, we make a detailed summary of the predicted distribution of each class, as shown in Table 2. EC1-EC6 denote the categories of enzymes. The values in the table indicate the probability that the enzymes in abscissa are predicted to be the category ordinate belongs to. In the horizontal comparison of Table 2, the prediction accuracy of each enzymatic class in Architecture 2 is higher than that in Architecture 1; at the same time, the longitudinal comparison shows that the accuracy of the proposed model ADL-DCNN for each enzymatic class is better than that of the model AD-CNN, regardless of the architecture.

In the training process, each iteration will produce different parameters, once training is completed, the corresponding parameters will be determined, then the accuracy is obtained by verification of test set. Usually, the precision of training set is always higher than that of test set, and with the number of iterations increasing, the precision gradually increases to convergence, but the model error reduces until there is no change, as shown in Figure 1 and Figure 2. Both accuracy and error can evaluate the performance of the model.

We use receiver operating characteristic (ROC) curves and area-under-the-curve (AUC) values to further assess the performance of the DCNN model, as shown in Figure 3. The assessment method is performed based on the final decision scores in classification scheme. The ROC curve is produced by the cross-validation experiments. Figure 3 shows ROC curve and AUC value of each class. The ordinate represents the true positive rate and the abscissa denotes false positive rate.

## 3. Method

In this section we describe the method used in the experiment. Firstly we will extract the feature of amino acid sequence in enzymes, and retaining the position and mutation information in the sequence is the primary focus of our work. Then, we will briefly describe the extract of structural features in proteins. In order to use three features simultaneously to predict protein function, a model framework with three parallel CNN is established.

### 3.1. Feature Extraction Based on Amino Acid Sequence

#### 3.1.1. Extracting Sequence Information by Position Specific Scoring Matrix

Position Specific Scoring Matrix (PSSM) is generated using PSI-BLAST [39]. Elements of this matrix represent the probability of mutating of amino acid at each position. Hence, PSSM is considered as a measure of residue conservatism in a given location.

The mutation information from PSSM is stored in a matrix, and formulated as
(1)$$PSSM={\left[\begin{array}{cccc} p_{1,1} & p_{1,2} & \cdots & p_{1,20}\\ p_{2,1} & p_{2,2} & \cdots & p_{2,20}\\ \vdots & \ddots & \vdots & \vdots \\ p_{L,1} & p_{L,2} & \cdots & p_{L,20} \end{array}\right]}_{L\times 20}$$
where, assume in an arbitrary protein, each amino acid $a_{i},i=1$, ⋯, *L*, *L* is the length of amino acid sequence, for arbitrary amino acid $a_{j},j=1$, ⋯, 20, 20 denotes the 20 native amino acids. The probability that $a_{i}$ mutates into $a_{j}$ is $p_{i}{,}_{j}$. Each $p_{i}{,}_{j}$ in PSSM is calculated as
(2)$$P_{i}{,}_{j}=\sum_{k=1}^{20}\gamma \left(i,k\right)\times w\left(k,j\right)\;\left(i=1,\cdots ,L;j=1,\cdots ,20\right)$$
where, γ(i,k) is the ratio between the frequency of appearing the k-th amino acid (among the 20 amino acids) at the position *i* of the probe and total number of probes, w(k,j) is the value of about the element in Dayhoff’s mutation matrix, which describes the rate that k-th amino acid type in a protein sequence change to *j*-th amino acid type with time elapsing. In PSSM, small value of each element indicates that there is less conservatism in the corresponding areas, whereas large values indicate quite conservative zones.

#### 3.1.2. Standardizing PSSM with 2D Histogram and Feature Visualization

In our application, we cannot convert the PSSM directly to feature vectors as proteins have amino acid sequences of different lengths, which will lead to different sizes of feature vectors, so we use 2D histograms to normalize feature vectors. The mutation feature maps of 2D histogram are produced as follow:For each type amino acid in PSSM, we extract the conservative probability of their different positions of the peptide chain, and the mutating information of each type amino acid is stored respectively as a matrix $Y_{nj}\;\left(n,j=1,\cdots ,20\right)$.The column of matrix $Y_{nj}$ was extracted using equally sized bins. Due to $p_{i,j}\in \left[-12,13\right]$, namely, the number of histogram bins is 25, a dimension [20 × 25] matrix is formed.The sequence of a protein feature maps $X_{L}$ (as shown in Figure 4) are achieved by the processing, and the dimension of $X_{L}$ is $\left[s\times s\times h_{L}\right]$, where, s=20 is the number of amino acids, $h_{L}=25$ is also equivalent to the number of channels. Finally, the histograms are smoothed by moving average filtering with a 1D Gaussian kernel (σ=0.5).

In Figure 4, the vertical axis at each plot corresponds to the 20 amino acids sorted alphabetically from left to right. The horizontal axis at each plot represents the mutation values in the range [−12, 13], and the bigger the value is, the greater the mutability is. The color means the mutation frequency that the amino acid in peptide chain mutates into an amino acid in a specific range.

In the process of deep learning training, because protein features as input data cannot be intuitively seen, we show each extracted feature maps by visualization.

Figure 4 shows the feature maps $X_{L}$ of amino acid mutation information in the sample protein, and the feature maps express the probability distribution($X_{L}$) of 20 amino acids mutating to other amino acids. As can be seen from the figure, the feature maps indicate clear differences between samples of different amino acids. In the figure, the darker grid indicates that the amino acid mutate to a certain amino acid more frequently, while the lighter indicates that the mutation frequency is less. It can be noticed that for all amino acids, most mutation probability are concentrated in the middle bins, corresponding to more probability values distributed over that range among the mutation of amino acids. Overall the spatial patterns in each class are distinctive and form uniquely multi-dimensional features for each protein sample.

### 3.2. Feature Extraction and Visualization Based on Protein 3D Structure

The 3D structure of the protein is expressed by the two torsion angles of the polypeptide chain and the distance between pairwise amino acid.

For each amino acid, the torsion angles are ϕ(N−Cα ) and ψ(Cα−C), the density of the torsion angles was estimated with equally sized bins based on the 2D sample histogram, and smoothed with a 2D Gaussian kernel [33]. The obtained feature maps $X_{A}$ of the angles are $\left[h_{A}\times h_{A}\times m\right]$, where, m=23 is the number of channels, $h_{A}=19$ is the number of bins.

In the proteins, for two arbitrary amino acids $a_{i}$, $a_{j}$, i,j=1, ⋯, *m*, the distance $d_{ij}$ between $a_{i}$ and $a_{j}$ is computed based on coordinates of the Cα atoms for the residues. Since the length of $d_{ij}$ is different in the proteins, the sample histogram of $d_{ij}$ is extracted to standardize measurements, the feature maps $X_{D}$ of the distance $\left[m\times m\times h_{D}\right]$ are achieved by convolution with a Gaussian kernel, and the range of distances is [5,40], where $h_{D}=8$ is the number of histogram bins (the number of channels) [33].

The structural information of protein is represented by the angle and the distance of amino acids. In Figure 5, three amino acids (ASE, PHE, THP) are taken as an example. Figure 5 shows 2D histogram of torsion angles ($X_{A}$) for each amino acid. The angular distribution of feature map is showed in Figure 5 for the three randomly selected amino acid. Examples of features maps representing amino acid distances ($X_{D}$) are illustrated in Figure 6, and the range of the vertical axes at each plot is [5,40]. Figure 6 has been produced by selecting the same amino acids as in Figure 5 for ease of comparison of the different feature representations. It can be seen that the distances between the amino acids in the title and 23 amino acids are concentrated in the last bin, corresponding to high distances.

In Figure 5, the horizontal and vertical axes at each plot correspond to and angles and vary from $-180^{\circ }$ (top left) to $180^{\circ }$ (right bottom). For arbitrary amino acid, navy blue indicates a large number of occurrences of the specific value $\left(\phi ,\varphi \right)$

In Figure 6, the horizontal axis at each plot represents the 23 amino acids sorted alphabetically from left to right. The vertical axis at each plot corresponds to the histogram bins. Each column shows the histogram of distances for an amino acid in the title and the one corresponding to the column.

### 3.3. Feature Extraction and Classification Based on Three Parallel CNNS

In this section, we introduce our framework as Figure 7 shown for predicting enzyme functions from deep CNN, and the model is composed of the following modules: convolutional Layer, batch normalization, rectified linear unit (ReLU), dropout and max pooling Layer, fully connected layer and softmax. In the framework, the CNN performance can be measured by a loss function that assigns a penalty to classification errors. The softmax loss function is used to predict the probability distribution over categories. After softmax normalization, the network outputs are used as class probabilities.

In Figure 7, the framework describes two processes: (1) three multi-channel feature maps are introduced respectively to a CNN; (2) After three feature maps are completed, the fused training results are input to a KNN classifier.

We adopt two fusion strategies: Architecture 1 and Architecture 2. In Architecture 1, angle feature map $X_{A}$, distance feature map $X_{D}$ and mutation feature map $X_{L}$ are each introduced into a CNN, which performs convolution at all channels, and then the class probabilities produced for each feature map are combined into a feature vector of length l∗3; In Architecture 2, each one of the ($m=23,\;h_{D}=8\;or\;h_{L}=25$) channels of each feature map is introduced independently into a CNN and the obtained class probabilities are concatenated into a vector of 1∗m features for $X_{A}$, $1*h_{D}$ features for $X_{D}$ and $1*h_{L}$ features for $X_{L}$, respectively. Three feature vectors are further combined into a single vector of length $l*(m+h_{D}+h_{L}=336)$.

In final classification, KNN(k=12) is applied for class prediction using as distance measure between three feature vectors $x_{A}\in X_{A}$, $x_{D}\in X_{D}$ and $x_{L}\in X_{L}$, and the metric is as follows $1-cov(x_{A},x_{D},x_{L})$, Where, measures the correlation between $x_{A}$, $x_{D}$ and $x_{L}$.

## 4. Discussion

This method has been applied for the prediction of the primary EC number and achieved 92.34% accuracy, which is a considerable improvement over the accuracy obtained in previous work (90.1% in Evangelia I [33]) when adding sequence information was incorporated. More particularly, except enzyme EC3, the prediction precisions of all enzymes in the model of this paper are higher than that in the model AD-CNN. For enzyme EC3, the the reason of slightly lower precision is that there may be the small number of these enzymes in the dataset. If the amount of data increases, the reliability on predictions can be improved.Validation and optimization of the model would be necessary to improve performance and provide better insight into the capabilities of this method.

When our experiment was completed, we have invited biological related personnel to test our model. They affirmed the prediction results of our model and thought that the model played an important role in prediction of biological protein function in general. Our approach can be used to discover hidden information when the location information of amino acid sequence changes. therefore, our finding serves as an innovative approach for future researchers who can utilize biological homology to increase performance results.

We believe that our current work demonstrates the effects of CNN models and PSSM profiles on protein function classification to outperform traditional methods. Nonetheless, we still face some limitations in this paper. Firstly, for the classification of enzyme function, 6 enzyme categories were roughly given, and we did not partition their functional detailedly. In the future, we may improve the model to adapt to more specific protein functions. Then, we should use the original 3D topological structure rather than the extraction of any statistical features as a learning image to investigate the predictive power. In short, the next work we will improve the performance of the model to speed up the training because the training of the model takes a lot of time throughout the experimental process.

## 5. Conclusions

In this paper, a method that extracts sequence feature and structural feature was presented, and we constructed a three-parallel deep CNN model to learn extracted features. The experimental results showed that the combination of sequence information and structural information led to more accurate prediction than the individual structural. Overall, the presented approach can provide a quick and accurate enzyme function prediction, and the research based not only structure but protein sequence for function prediction provides the more comprehensive molecular representation of biological field. In future work, we will try to apply the method to prediction of the hierarchical relation of function subcategories and annotation of enzymes.

## Figures and Tables

**Figure 1 ijms-20-02845-f001:**
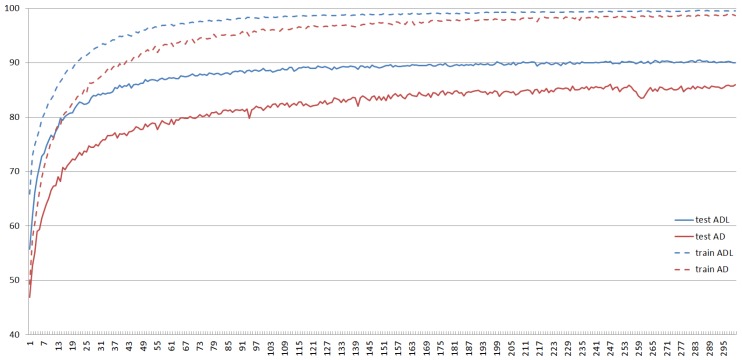
The precision change of two different models. The two red curves show training and test accuracy of the model AD-CNN change with the number of iterations varying. Likewise, the two blue curves indicate two precisions of the model in this paper vary with the number of iterations changing.

**Figure 2 ijms-20-02845-f002:**
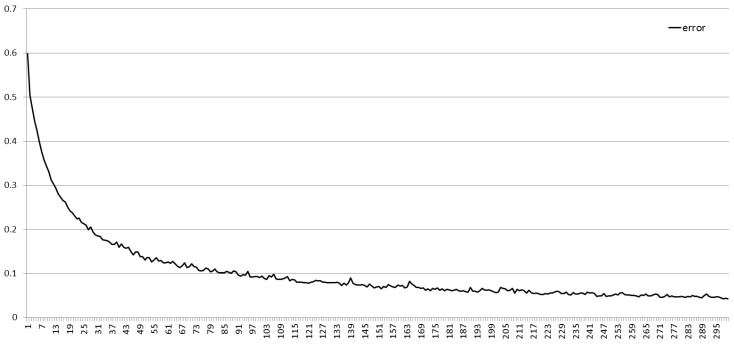
Error change of different the number of iterations.

**Figure 3 ijms-20-02845-f003:**
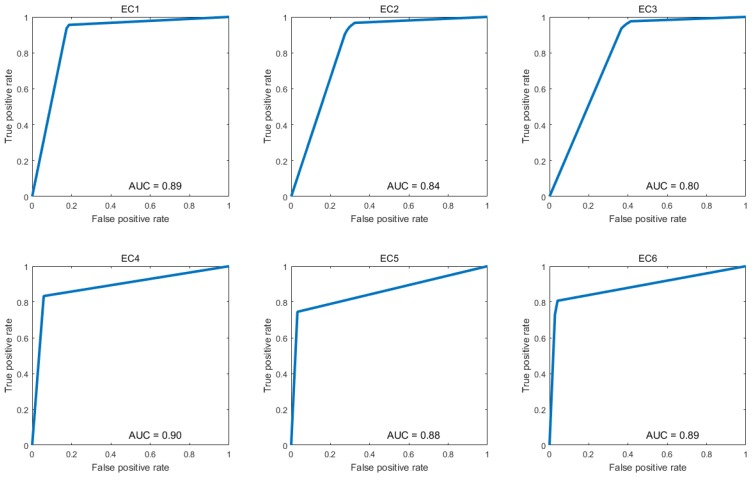
ROC curves for each enzymatic class for Architecture 2.

**Figure 4 ijms-20-02845-f004:**
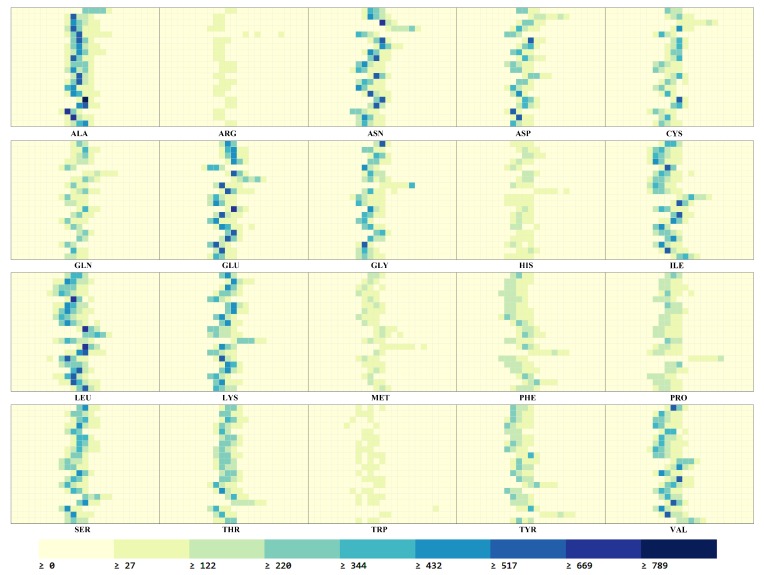
Feature maps of peptide chain mutation information.

**Figure 5 ijms-20-02845-f005:**
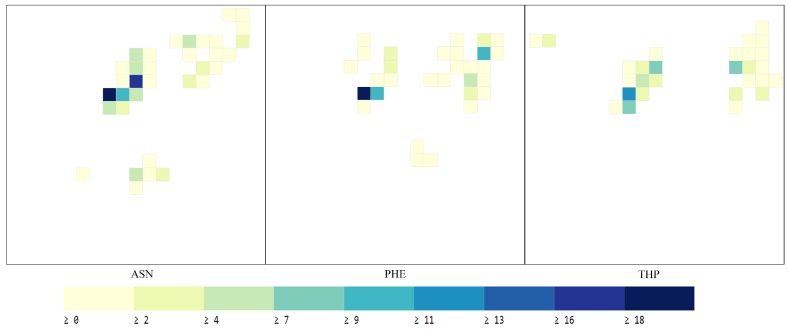
Torsion angles feature maps.

**Figure 6 ijms-20-02845-f006:**
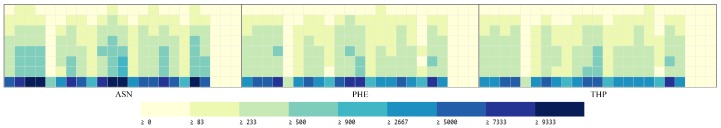
Feature maps of paiwise amino acid distances.

**Figure 7 ijms-20-02845-f007:**
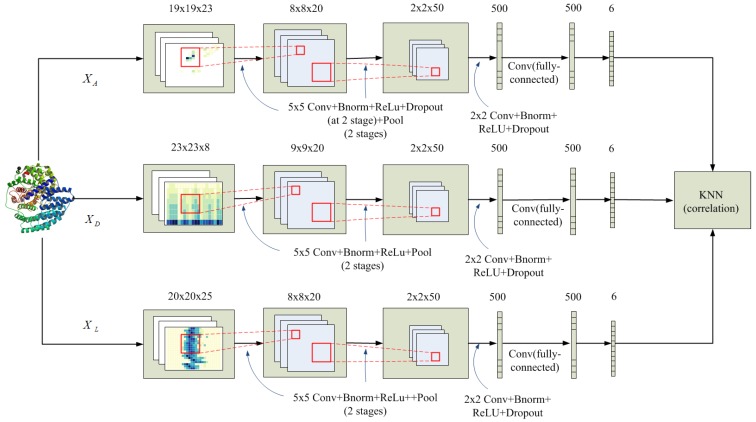
Framework of the DCNN.

**Table 1 ijms-20-02845-t001:** Prediction accuracy of enzymatic function in different models with two architectures.

		Architecture 1		Architecture 2	
Class	Sample	AD-CNN Acc(%)	ADL-DCNN Acc(%)	AD-CNN Acc(%)	ADL-DCNN Acc(%)
EC1	16,669	92.05	93.11	94.32	95.86
EC2	1893	61.73	74.23	75	74.23
EC3	1757	57.42	73.95	72.55	74.51
EC4	3102	75.44	81.80	81.98	83.25
EC5	7968	87.61	92.26	92.01	93.84
EC6	7968	87.41	90.02	92.79	94.39
Total	43,843	85.97	90.01	90.83	92.34

**Table 2 ijms-20-02845-t002:** Matrices for each fusion scheme and feature maps.

		Prediction by Architecture 1						Prediction by Architecture 2					
Feature Maps	Class	EC1 Acc(%)	EC2 Acc(%)	EC3 Acc(%)	EC4 Acc(%)	EC5 Acc(%)	EC6 Acc(%)	EC1 Acc(%)	EC2 Acc(%)	EC3 Acc(%)	EC4 Acc(%)	EC5 Acc(%)	EC6 Acc(%)
AD-CNN	EC1	92.05	0.33	0.54	0.39	1.60	5.08	94.32	0.03	0.15	0.18	1.30	4.02
	EC2	10.46	61.73	1.79	2.30	6.63	17.09	9.95	75.00	0.26	1.02	4.34	9.44
	EC3	14.57	0.84	57.42	0.28	6.16	20.73	10.92	0.00	72.55	0.56	3.92	12.04
	EC4	8.77	0.80	0.64	75.44	4.94	9.41	5.26	0.16	0.32	81.98	4.94	7.34
	EC5	5.53	0.57	0.63	0.69	87.61	4.97	4.15	0.00	0.06	0.19	92.01	3.58
	EC6	8.06	0.44	0.44	1.08	2.57	87.41	4.81	0.16	0.20	0.28	1.76	92.79
ADL-DCNN	EC1	93.11	0.21	0.36	0.45	1.12	4.69	95.86	0.00	0.09	0.03	0.73	3.30
	EC2	7.65	74.23	2.81	4.85	1.60	8.67	9.95	74.23	0.77	0.77	4.08	10.20
	EC3	7.84	0.28	73.95	1.12	1.96	14.85	10.08	0.00	74.51	0.00	2.52	13.00
	EC4	6.86	1.28	0.48	81.80	3.03	7.18	4.63	0.16	0.16	83.25	3.83	7.97
	EC5	3.71	0.53	0.19	0.50	92.26	2.83	3.40	0.00	0.00	0.06	93.84	2.70
	EC6	6.75	0.40	0.64	0.60	1.68	90.02	4.53	0.04	0.08	0.12	0.84	94.39

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
