# Peer review of "Prediction of Enzyme Function Based on Three Parallel Deep CNN and Amino Acid Mutation"

_ijms, 2019, doi:10.3390/ijms20112845_

Round 1
Reviewer 1 Report
1. There are many grammatical and typos errors in this manuscript. The authors need to re-check and revise them carefully. Some examples are as follows:
... probability of the amino acids,and structure ...
... that model to classify a number of function categories for an input sequence ...
... deep learning which uses multiple layers representation and abstraction of data has drastically improved the accuracy ...
... proposed a deep learning based on the position-specific scoring matrix profile ...
...
2. I have checked and saw that there are some overlaps between this work and 2 published works:
Zacharaki, E. (2017). Computational methods towards image-based biomarkers and beyond (Doctoral dissertation, Université Paris-Est).
Li, L., Kong, L., Xie, B., Fang, X., Kong, W., Liu, X., ... & Zhao, F. (2019). The Influence on Response of a Combined Capacitance Sensor in Horizontal Oil–Water Two-Phase Flow. Applied Sciences, 9(2), 346.
3. Please rephrase the sentences and cite these 2 works to avoid plagiarism errors.
4. In the literature review, the authors did not mention the previous works which focused detail on enzyme classification. They only talked about protein function prediction.
5. To mention protein function prediction, there is a need to provide more latest references such as https://doi.org/10.1186/s12859-016-1163-x and https://doi.org/10.1002/jcc.24842.
6. In some references, the authors used "al et." instead of "et al."
7. How did the authors collect enzymes from PDB? For example, what is the keyword/query? Also, please include the time that they collected data.
8. In most of the similar works in this field, researchers tried to remove similarity sequences before performing experiments. I think it is a very important step and the authors have to follow this rule. Without this step, the experiment results are not reliable.
9. Table 2 is very confusing. Because the confusion matrix shows the number of TP, FP, TN, and FN, but here are not integers.
10. Figure 1 showed training or validation accuracy? There is a need to provide both of them in the same figure.
11. The authors should mention that PSSM profiles had been widely used in a lot of problems in this field, such as https://doi.org/10.1016/j.ab.2018.06.011, https://doi.org/10.1186/s12859-016-1369-y, and https://doi.org/10.1016/j.jmgm.2017.01.003.
12. Similar, applying CNN on PSSM profiles had been successfully used in https://doi.org/10.7717/peerj-cs.177 and https://doi.org/10.1016/j.ab.2019.03.017, but the authors missed to discuss these published works to see the differences and their contribution.
13. The authors have to create a web server to facilitate more users to assess the proposed method.
14. The authors have to compare their proposed method with the other published works on the same dataset.
Author Response
1. There are many grammatical and typos errors in this manuscript. The authors need to re-check and revise them carefully. Some examples are as follows:
(1)... probability of the amino acids, and structure ...
Response (1): This sentence has been modified in line 7-line8.
(2)... that model to classify a number of function categories for an input sequence ...
Response (2): This sentence has been modified in line 48-line49.
(3)... deep learning which uses multiple layers representation and abstraction of data has drastically improved the accuracy ...
Response (3): This sentence has been modified in line 53-line54.
(4)... proposed a deep learning based on the position-specific scoring matrix profile ...
Response (4): This sentence has been modified in line 58.
Response 1: We have carefully checked our papers and other errors have been fixed.
2. I have checked and saw that there are some overlaps between this work and 2 published works:
Zacharaki, E. (2017). Computational methods towards image-based biomarkers and beyond (Doctoral dissertation, Université Paris-Est).
Li, L., Kong, L., Xie, B., Fang, X., Kong, W., Liu, X., ... & Zhao, F. (2019). The Influence on Response of a Combined Capacitance Sensor in Horizontal Oil–Water Two-Phase Flow. Applied Sciences, 9(2), 346.
Response 2: We have read these references carefully and modified some overlaps.
3. Please rephrase the sentences and cite these 2 works to avoid plagiarism errors.
Response 3: References has been cited in line 68.
4. In the literature review, the authors did not mention the previous works which focused detail on enzyme classification. They only talked about protein function prediction.
Response 4: We have added some references of protein classification that contains enzyme classification.
5. To mention protein function prediction, there is a need to provide more latest references such as https://doi.org/10.1186/s12859-016-1163-x and https://doi.org/10.1002/jcc.24842.
Response 5: According to your opinion, we read these references carefully and found that the views in the text are worth learning, so we have added these references to the relevant description in line 30 and line 31.
6. In some references, the authors used "al et." instead of "et al."
Response 6: We have checked and modified the problems.
7. How did the authors collect enzymes from PDB? For example, what is the keyword/query? Also, please include the time that they collected data.
Response 7: The data set about proteins (n=43843) was retrieved from the PDB (http://www.rcsb.org/) database. The keywords are the enzyme IDs from references [33].
8. In most of the similar works in this field, researchers tried to remove similarity sequences before performing experiments. I think it is a very important step and the authors have to follow this rule. Without this step, the experiment results are not reliable.
Response 8: The similarity of amino acid sequences greatly affects the function of proteins. Early, for protein function prediction, only amino acid sequence was used to predict protein function. Because the protein function is not only related to the protein sequence, but also related to the secondary structure of the protein, the prediction accuracy could not be improved very well when the data set was limited. At present, there is not a good model to achieve a high prediction accuracy, but when the structural information of the protein is added to the model, the prediction result is much better than the sequence information only. The protein function may also be different when the amino acid sequences have high similarity. In this paper, we add the structural features of the protein on the basis of the sequence, so the effect of sequence similarity on protein function is reduced.
9. Table 2 is very confusing. Because the confusion matrix shows the number of TP, FP, TN, and FN, but here are not integers.
Response 9: The values in Table 2 indicate the probability values corresponding to TP, FP, TN, and FN. In addition, we also made a roc curve corresponding to the TP, FP, TN, and FN values to represent the model performance.
10. Figure 1 showed training or validation accuracy? There is a need to provide both of them in the same figure.
Response 10: Figure 1 is modified, in the figure, the solid line represents the change in the accuracy of the test set data, and the dotted line represents the change in the training set data.
11. The authors should mention that PSSM profiles had been widely used in a lot of problems in this field, such as https://doi.org/10.1016/j.ab.2018.06.011, https://doi.org/10.1186/s12859-016-1369-y, and https://doi.org/10.1016/j.jmgm.2017.01.003.
Response 11: We have included some related applications of PSSM in line 29-39 and cited these references in References [7] [8] [9].
12. Similar, applying CNN on PSSM profiles had been successfully used in https://doi.org/10.7717/peerj-cs.177 and https://doi.org/10.1016/j.ab.2019.03.017, but the authors missed to discuss these published works to see the differences and their contribution.
Response 12: We have cited these references in References [10][34], and discussed their contributions and the differences in this field.
13. The authors have to create a web server to facilitate more users to assess the proposed method.
Response 13: In general, we really need a web server to allow others to evaluate our methods. Since the research lacks related personnel that are proficient in web server, we have invited some experts that learn about machine learning to do a small amount of testing, which proved that our method had a certain effect on the prediction of protein biological data.
14. The authors have to compare their proposed method with the other published works on the same dataset.
Response 14: We have shown the compared results of our method and the previous method under the same data set in Tables 1 and 2 that have been modified.
Reviewer 2 Report
Major points:
1. Did the author’s select different chains of the same PDB for their study. If yes, then their training has the chance to overperform the prediction. No information was provided in the manuscript. Need to clearly justify this point.
2. The assessment of model error needs to be added in the results to get a deeper insight into their method performance
3. Not many details are provided about the data collection and the pre-process step. This is very important for the model performance and usability of the developed method.
Minor points:
1) In the introduction, line 17-19 can be separated to make clear for the audience.
2) Line 25, and line 26 reference needs to be quoted. Because this is the main focus of the current study and needs to be properly cited.
Author Response
1. Did the author’s select different chains of the same PDB for their study. If yes, then their training has the chance to overperform the prediction. No information was provided in the manuscript. Need to clearly justify this point.
Response 1: The data set about proteins (n=43843) was retrieved from the PDB (http://www.rcsb.org/) database. The proteins used were enzymes of single class, whereas enzymes that perform multiple reactions and are associated with multiple enzymatic functions were excluded. What's more, the enzymes of small sequence length or homology bias were not considered. In the pre-process step of data, we first read the amino acid sequences in each protein pdb file on-line, then the scoring matrix of the protein was derived by PSI-BLAST.
2. The assessment of model error needs to be added in the results to get a deeper insight into their method performance
Response 2: We have added Figure 2 to show the variation of the model's error with the number of iterations changing.
3. Not many details are provided about the data collection and the pre-process step. This is very important for the model performance and usability of the developed method.
Response 3: The data collection and the pre-process are described in the experimental part of the article (line 93- 99).
1) In the introduction, line 17-19 can be separated to make clear for the audience.
Response 1): We have made corresponding modification in the text.
2) Line 25, and line 26 reference needs to be quoted. Because this is the main focus of the current study and needs to be properly cited
Response 2): We have made corresponding modification in the text.
Reviewer 3 Report
The authors have analyzed the pdb database and aimed to predict the function of enzymes based on the information extracted. First of all, the database could contain the proteins that do not express enzymatic activity. Authors should remove them of reconsider the title.
Next, the dataset could contain the same proteins with different ligands. Presence of such entries could result in bias of models. Moreover, it worth checking if the classification still accurate when only one protein is present in the data set (i.e. same proteins with different mutations are not in the same data set).
Deeper literature analysis is required. There are plenty of works that use CNN to represent proteins for different purposes.
It is unclear why authors decide on certain tuning options for the model. Did they try to develop some sort of baseline model first?
Are XA, XD, and XL different training/test splits? If not, what are they? More details are needed.
What were the accuracy measures uses in Table 1? This should be explicitly stated. Same goes for the Table 2.
The feature visualization technology should be presented in a clearer way.
Discussion is way too short.
Have the authors compared their performance with the performance of other techniques? It should be done, unless it is not clear if their methods is beneficial or not.
Author Response
1. The authors have analyzed the pdb database and aimed to predict the function of enzymes based on the information extracted. First of all, the database could contain the proteins that do not express enzymatic activity. Authors should remove them of reconsider the title.
Response 1: The data we use are all enzymes, and all pdb files of enzymes were read from PDB database, so the database didn’t contain proteins that do not express the enzymatic activity.
2. Next, the dataset could contain the same proteins with different ligands. Presence of such entries could result in bias of models. Moreover, it worth checking if the classification still accurate when only one protein is present in the data set (i.e. same proteins with different mutations are not in the same data set).
Response 2: 1) When proteins combine with different ligands, they express different functions. But in our dataset, the classification of enzymes is divided by biological professionals, and the enzymes are generally classified into six classes according to their functions, such as oxidoreductases (EC1), transferases(EC2), hydrolases (EC3), lyases (EC4), isomerases (EC5), ligases (EC6).
2) Analysis of proteins and mutant variants can indeed be used as an indicator to measure model fitness. Due to the high number and variety of proteins, our dataset cannot cover all kinds of mutations in proteins, but may contain other conditions like ligand binding. We mainly study the aspect of enzyme classification. The data set contains some variants of the protein and proteins that do not contain their variants, but we did not consider it from that level. On the contrary, we continue the work of our predecessors and measure the mutation of the protein with the overall accuracy. Hopefully this model is used for most enzyme classification problems. Your idea is a good suggest for the study of protein function changes after amino acid mutations. We will conduct further research in the future.
3. Deeper literature analysis is required. There are plenty of works that use CNN to represent proteins for different purposes.
Response 3: We have added an analysis of these works in introduction.
4. It is unclear why authors decide on certain tuning options for the model. Did they try to develop some sort of baseline model first?
Response 4: We made corresponding improvements based on this model that Evangelia et al [33] proposed in 2017.
5. Are XA, XD, and XL different training/test splits? If not, what are they? More details are needed.
Response 5: For each training step, we use the same batch of protein corresponding XA, XD, and XL feature maps, which are simultaneously input to the modules for training.
6. What were the accuracy measures uses in Table 1? This should be explicitly stated. Same goes for the Table 2.
Response 6: The accuracy in Table 1 refers to the proportion of each enzyme class correctly predicted, and the last row represents the overall prediction result; the accuracy values in Table 2 represent the proportion that the enzyme is predicted to be other enzyme classes.
7. The feature visualization technology should be presented in a clearer way. Discussion is way too short.
Response 7: It may be that the structure of the experimental method is not clear in this paper. We have modified the structure of the article to express the process of visualizing the feature map more clearly.
8. Have the authors compared their performance with the performance of other techniques? It should be done, unless it is not clear if their methods is beneficial or not.
Response 8: We show the compared results of our method and the previous method under the same data set in Tables 1 and 2 that have been modified.
Round 2
Reviewer 1 Report
The authors have addressed most of my comments. It can be accepted for publication in IJMS. To make a final version better, English language and style should be improved a little bit before production.
Author Response
The authors have addressed most of my comments. It can be accepted for publication in IJMS. To make a final version better, English language and style should be improved a little bit before production. To make a final version better, English language and style should be improved a little bit before production.
Response: We examined the manuscript carefully and found many mistakes, including spelling, omission and grammar. So far, all the mistakes have been modified, and we have asked the professional English staff to make the revision.
Reviewer 2 Report
The authors properly addressed the comments and it looks clear now.
Author Response
The authors properly addressed the comments and it looks clear now.
Response: We have still carefully checked the manuscript and modified several sentences.
Reviewer 3 Report
The authors have addressed the majority of the questions I raised. The level of English has improved, but minor spellcheck is still required.
Author Response
The authors have addressed the majority of the questions I raised. The level of English has improved, but minor spellcheck is still required.
Response: We examined the manuscript carefully and found many mistakes, e.g. in line 7, Line 96, Line 125, Line 132—138, Line 147, Line 155…., all the mistakes have been modified, and we have asked the professional English staff to make the revision.